# "Fear of the unknown": Health, disability, and stakeholder perspectives on the behavioral and social drivers of vaccination in children with disability in Fiji

**Rosalie Power**[1‡], **Unise Vakaloloma**[2‡], **Israt Jahan**[3], **Sureni Perera**[4], **Ilisapeci Tuibeqa**[5], **Rachel Devi**[6], **Litiana Volavola**[6], **William May**[7], **Donald Wilson**[2], **Lanieta Tuimabu**[8], **Gulam Khandaker**[9], **Meru Sheel**[10,11]*, on behalf of Australian Immunisation and Disability Investigators[¶]

1 Translational Health Research Institute, Western Sydney University, New South Wales, Australia, 2 Fiji Institute of Pacific Health Research, College of Medicine, Nursing & Health Sciences, Fiji National University, Suva, Fiji, 3 Central Queensland University, Queensland, Australia, 4 Frank Hilton Organization, Suva, Fiji, 5 Colonial War Memorial Hospital, Ministry of Health and Medical Services, Suva, Fiji, 6 Family Health, Ministry of Health and Medical Services, Suva, Fiji, 7 School of Medical Sciences, College of Medicine, Nursing & Health Sciences, Fiji National University, Suva, Fiji, 8 Fiji Disabled People's Federation, Suva, Fiji, 9 Central Queensland Public Health Unit, Central Queensland Hospital and Health Service, Queensland, Australia, 10 Sydney School of Public Health, The University of Sydney, New South Wales, Australia, 11 Sydney Infectious Diseases Institute, Faculty of Medicine and Health, The University of Health, New South Wales, Australia

‡ RP and UV share first authorship on this work.
¶ Membership of the Australian Immunisation and Disability Investigators is provided in the Acknowledgments.
* meru.sheel@sydney.edu.au

**Data Availability Statement:** The data supporting the findings of this study are qualitative in nature and contain potentially identifying information. To

## Abstract

Vaccinating children with disability in low- and middle-income countries, such as Fiji, is a key priority for equity. This study aimed to understand the social and behavioral drivers of vaccine uptake among children with disability in Fiji, from the perspectives of health, disability, and community stakeholders. Five qualitative focus groups were conducted with 22 stakeholders, including healthcare workers, disability service providers and advocates, and community and faith leaders (female $n = 17$, 77%). Data were collected and analyzed using reflexive thematic analysis and applied the World Health Organization's Behavioral and Social Drivers of Vaccination framework. Numerous drivers were reported to impact vaccination for children with disability. These included (1) *Thinking and feeling*: lack of reliable information about vaccine benefits and safety for children with disability; (2) *Social processes*: disability stigma and discrimination impacted families of children with disability engaging with healthcare services; lack of tailored vaccination communication and engagement strategies; and, need for improved disability and health service collaboration; (3) *Motivation*: lack of awareness and support for parents of children with disability to have their children vaccinated, and religious beliefs negatively impacted motivation; (4) *Practical issues*: long waiting times and lack of suitable waiting areas for children with disability; financial and time barriers; and, lack of healthcare worker knowledge and confidence in providing vaccines to children with disability, impacted patient-provider trust. The findings from this

protect the privacy of the participants, the full dataset will not be made publicly available. However, relevant excerpts of the data are published within the manuscript. Interested researchers may request access to the data by contacting the corresponding author directly, and access may be granted on a case-by-case basis, subject to ethical considerations and participant confidentiality agreements.

**Funding:** This study was funded by the Australian Government Department of Foreign Affairs and Trade (G207535) under the Australian Regional Immunisation Alliance - Regional Immunisation Support and Engagement (ARIA-RISE) scheme. IJ is supported by the early career researcher grant of the Cerebral Palsy Alliance Research Foundation (ERG02021 and ERG05123). GK is supported by the NHMRC Investigator Grant (APP2009873). MS was supported by a Westpac Research Fellowship in 2022-2023. The funders had no role in study design, data collection and analysis, decision to publish, or preparation of the manuscript.

**Competing interests:** The authors have declared that no competing interests exist.

study can inform strategic actions to overcome barriers to vaccination for children with disability, including strengthening existing vaccination programs, promoting greater equity in vaccination for children with disability in Fiji. This will reduce the burden of vaccine-preventable diseases in this priority group.

## Introduction

Vaccination is one of the most successful public health interventions to reduce childhood morbidity and mortality on a global scale [1]. In recent decades [2], significant progress has been achieved through immunization programs, improving health outcomes for vulnerable populations [3] contributing to the significant decline in morbidity and premature mortality associated with vaccine-preventable diseases [4]. However, this progress is not evenly distributed, particularly among children with disability in low- and middle-income countries (LMICs) [5]. Over 90% of children with disability worldwide live in LMICs [6], highlighting the inequitable distribution of childhood disability. It is estimated that in Fiji, an upper-middle-income country [7] in the Western Pacific, approximately 10% of children have one or more impairments [8]. Although, this is likely to be an underestimate due to the poor availability of data on disability, stigma, and lack of infrastructure impacting the identification and diagnosis of children with disability [9].

For more than a decade, the Fiji Ministry of Health and Medical Services (MHMS) has demonstrated a steadfast commitment to improving childhood vaccination through collaborations with national and international partners [10, 11]. Nationwide vaccination campaigns, such as the introduction of rotavirus and 10-valent pneumococcal conjugate vaccine (PCV10) in 2012, have led to positive outcomes, including reduced disease incidence, hospitalization, and mortality from vaccine-preventable diseases among children under 59 months old [10, 12, 13]. However, the latest data from multiple indicator cluster survey (MICS) [8] showed that over one-third (36%) of children under 36 months were not vaccinated against basic antigens as per the vaccination schedule in 2021. Currently, there are no national immunization coverage data for children with disability for vaccines provided under the national immunization program or for COVID-19 vaccines. However, it is hypothesized that children with disability make up a significant portion of the unreached and unvaccinated population in Fiji. This is because children with disability often face multiple barriers to accessing vaccination due to the impacts of disability, poverty, and inequity in access to basic primary healthcare services [5]. Moreover, the COVID-19 pandemic disrupted routine immunization service delivery in most settings, eroding consecutive annual gains in immunization coverage, although is slowly recovering [14]. In Fiji, the coverage for DPT3 in 2022 dropped to 85.0% compared with 91.8% in 2019, with rates recovering to 95.2% in 2023 [15].

To better understand access to immunization among children with disability in Fiji and to inform the present study, we conducted a cross-sectional survey to examine vaccination rates in children with disability. We found that only 55% of children with disability in the study location (the same location as the present study) were vaccinated against basic antigens [16]. Key factors significantly associated with partial vaccination included a lack of confidence in vaccine safety and an inability to access vaccines [16]. It is well known that healthcare providers play a crucial role in addressing these barriers and influencing both decision-making and access to immunization for children with disability [5, 17]. Additionally, in Fiji, stakeholders outside of the healthcare system, including disability and community stakeholders (e.g.,

education providers and those in faith leadership roles), are instrumental in shaping public perceptions and attitudes to vaccination and influencing health behaviors for people with disability [9, 18, 19]. Therefore, to increase vaccination coverage, it is essential to understand the attitudes and examine the perspectives of multiple stakeholder about the vaccination of children with disability and the factors that enable and prevent vaccine uptake in this population [20] In this study, we examined *the social and behavioral drivers of vaccine uptake among children with disability in Fiji from the perspective of health, disability, and community stakeholders.*

## Materials and methods

### Design and theoretical framework

The study is a two-part study looking at immunization in children with disability. Part 1 involved a cross-sectional survey examining vaccine uptake among 198 children with disability and their caregivers (published elsewhere, see [16]). Part 2 involved a qualitative study utilizing focus groups with health, disability, and community stakeholders on the behavioral and social drivers of vaccination uptake among children with disability. In this paper, we present qualitative data from the focus groups.

A 12-member stakeholder group was actively engaged and involved throughout all stages of the project providing expertise to guide the study design, data collection, analysis, and dissemination of results [21, 22]. The group included local and international disability and health service providers (study partners), MHMS, academics, and community and faith leaders. The stakeholder group identified focus groups as the preferred method for data collection, as participants were anticipated to feel more comfortable sharing their opinions in a group setting, reducing the pressure of individual interviews or a survey, and enabling them to control their level of participation [23]. Additionally, the group setting fosters interaction among participants, encouraging the exchange of ideas and yielding a deeper, richer understanding of sensitive and potentially taboo topics [23].

We used the World Health Organization's (WHO) Behavioral and Social Drivers (BeSD) of Vaccination Framework [20] to provide a structured approach to understanding the complex factors influencing vaccination uptake among children with disability. This framework offered a comprehensive guide to examine not only individual beliefs and behaviors but also social, cultural, and systemic drivers that can impact vaccine acceptance, access, and uptake. This framework is well-suited for informing strategies to improve equitable vaccine uptake in contexts with diverse social and healthcare challenges [20].

### Study location

This study was conducted in peri-urban and urban areas of the Suva–Nausori corridor. The Suva-Nausori corridor is a 19km stretch between Suva (capital of Fiji) and Nausori on its north east. The study site was chosen in collaboration with Fiji's Ministry of Health and Medical Services and non-governmental partners to ensure representation from both rural and urban areas. Based on in-country unpublished data it is believed to be an area with high prevalence of functional impairments among children (6% in ages 2–4 and 9% in ages 5–17) [8], as well high population density (approximately 164,000) [24], and challenging access to routine healthcare services.

### Recruitment and participants

For the focus groups, we purposively recruited immunization health workers, disability service providers and advocates, and community and faith leaders from local non-government, international and Government agencies [25]. Community leaders included education providers of

children with disability and faith leaders. We used study investigator and partners' networks to identify the participants. Data collection occurred from 12th June 2023 to 1st August 2023. Potential participants were initially contacted by phone, followed by an email with the study information sheet and consent form, which was signed and returned to the research team. Participants were eligible for the study if they (a) worked in the Suva-Nausori corridor; and (b) provided health, disability, or community services to children aged 2 to 19 years old with disability. On the advice of our stakeholder group, participants working at the same organization were allocated to separate focus groups, while people of similar hierarchical positions were grouped together to manage power dynamics. This approach allowed participants to engage more candidly without concerns about authority or judgement from higher-ranking colleagues and minimized the risk of self-censorship and inhibited discussion [26].

In addition to the signed forms, consent was also confirmed again verbally before the focus group interviews by reading the consent form aloud to ensure clarity and answering any questions before commencing focus group discussions. The discussions had an average of four participants per session, ranging from 30 to 60 minutes.

## Procedure

Two focus group interview guides (one for immunization healthcare workers and another for disability and community stakeholders) were adapted from the WHO BeSD childhood vaccination qualitative framework analysis template tool (Annex 1, WHO BeSD Vaccination Framework [20]). The adaptation process was conducted in collaboration with the in-country investigator team and stakeholder group, followed by pilot testing with two groups to confirm feasibility. Modifications included incorporating local terminology, improving in-depth probing, and asking specifically about the vaccination of 'children with disability or special needs'. Additional prompts were introduced to evaluate immunization efforts targeting underprivileged or remote populations and outreach strategies such as door to door services, see S1 Appendix. One of the study investigators (RP) trained the lead interviewer (UV) on how to conduct and moderate focus group discussions. As participants were proficient in English focus groups were mainly conducted in English, with some Fijian words and phrases used from the Fijian version schedule to aid understanding as required.

At the start of each focus group, participants were again provided with the study information sheet, which was read aloud to support access and to facilitate understanding. The lead interviewer reminded participants of the study's aim and objectives, that participation was confidential, and that they could withdraw from the study until the interview transcripts had been de-identified.

During the focus groups, participants were asked about their roles in providing vaccination to children with disability, the processes they follow to immunize a child with disability, including navigating consent, drivers and barriers to vaccinations for this population, and how they thought immunization services could be improved for children with disability in their locale. To facilitate relaxed interaction between participants, the facilitator built rapport with participants before formally commencing the interviews [26]. The facilitator also used body language, such as nodding and hand gestures, to encourage participation among quieter members of each group and asked probing questions to facilitate group conversation [26]. Data were collected between June and August 2023.

## Data analysis

A combination of inductive and deductive reflexive thematic analysis was used to analyze the focus group discussions [27]. Within this framework, themes are generated through extended

engagement with the data, "a creative and active process" [27, p.343]. The interviews were transcribed verbatim and checked for accuracy by two researchers (LS & UV). The transcripts were then de-identified. To code the data, four researchers (RP, MS, IJ, and UV) each independently read a transcript line-by-line and made notes to capture concepts identified in the data. The researchers then met to compare similarities and differences across the concepts. Through an iterative process of collaborative discussion and decision-making, the categories were grouped and ordered where commonalities occurred. Differences of opinion were managed through discussion until a consensus was reached. The codes were refined, and definitions of what data should be included within each code were determined. Having formulated the coding framework, the transcripts were then imported into NVIVO July 2023 (Release 14.23.2), a software to facilitate the organization of qualitative data into the codes. Once the coding was complete, each of the code sections was summarized, which helped to identify similarities and facilitated the generation of themes and sub-themes. Through a further process of discussion and decision-making, the themes were refined and clarified, then organized using the WHO BeSD framework for understanding behavioral and social drivers of vaccination and its key domains: what people think and feel, social processes, motivation, and practical issues. Throughout the analysis, we actively engaged in reflexivity, being aware of our role in knowledge production. We critically reflected on how our cultural backgrounds, ages, genders, and other intersectional characteristics and experiences shaped our assumptions and influenced the research process.

### Ethical approval

Ethical approval was obtained from three institutes: the Fiji Human Health Research Ethics Committee (protocol number 05/2023); the Fiji National University College of Human Health Research Ethics Committee (protocol number 155.22), and the University of Sydney Human Research Ethics Committee (protocol number 2023/534). In line with local procedures, multiple health facility approvals were also obtained from the Permanent Secretary of Fiji, the Division of Medical Officers, and the Sub-divisional Medical Officers in the Central Division.

## Results

We conducted five focus groups with 22 participants. Most participants were female ($n = 17$, 77%). Participants represented three key groups: Healthcare workers (6, 27%), disability service providers [NGOs & International Agency] (9, 41%), and community and faith leaders (7, 32%).

The key drivers of vaccination were grouped according to the WHO BeSD vaccination framework (see Fig 1). Detailed results by domain are presented below.

### Thinking and feeling

**'Fear of the unknown': Concerns about vaccine benefits and safety.**    Participants reported "resistance from parents" [Male, Education stakeholder] on having their children with disability vaccinated due to "not having the knowledge on the importance of vaccination" [Female, Health worker]. Participants explained that that there was a lot of "misinformation" [Female, NGO] about vaccines among parents of children with disability, "because of plenty coconut wireless around, sharing information which is not correct" [Male, Education stakeholder]. This included concern among some parents that vaccines caused their child's disability, as a participant commented:

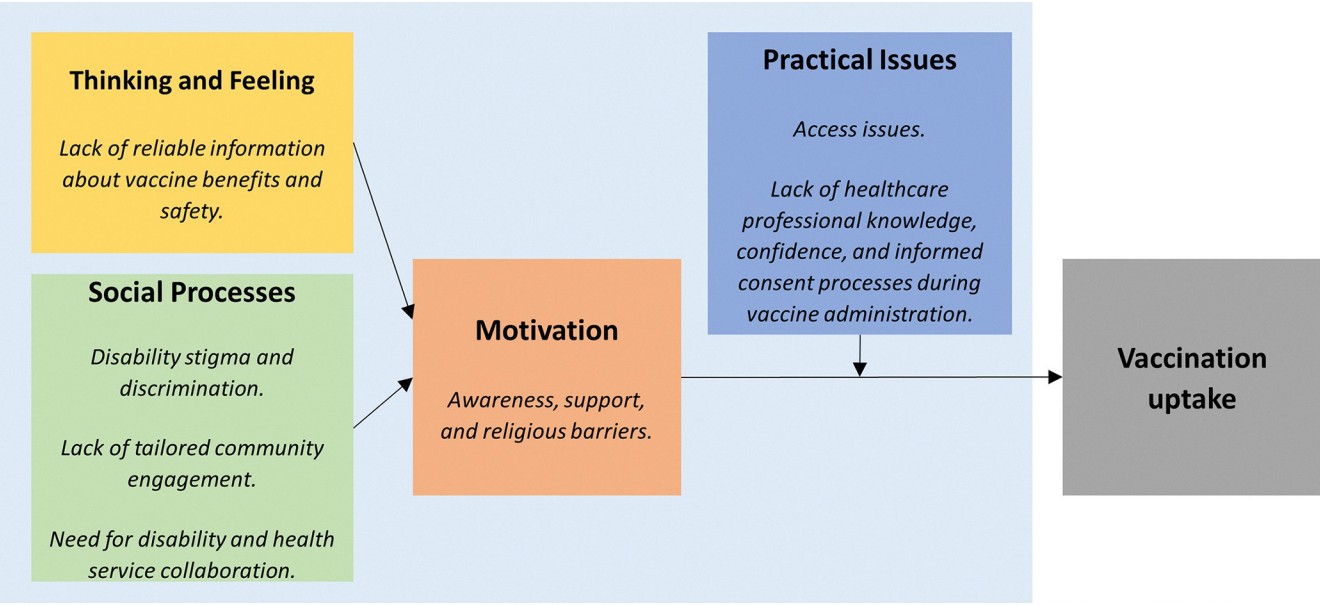

**Fig 1. Thematic findings grouped using the behavioral and social drivers of vaccination framework.**

We have parents coming in with their child with disability who say, "Oh, they might have gotten the wrong vaccination or the wrong injection, and that is why my child is this way or that way" [Female, Disability worker].

Participants reported that when information about vaccines was provided to parents, it was often incomplete focusing solely on the benefits and neglecting potential side effects. This was reported to cause distress for parents, as their children sometimes became unwell following the vaccination. A participant commented, "You know the phrase 'the fear of the unknown'. That's what happens when it is not explained well" [Female, Community/ faith leader]. Another participant said:

Sometimes when information about vaccination is shared, they only seem to just highlight the positives. It says, "This will protect you from so and so", it does not highlight the side effects, so everybody says "I'm going to be protected, I'm going to be immunized" but when they go, they return with a headache or weak bodies, they come and talk about it and those that have not been immunized, they don't want to go [Female, Disability worker].

Participants reported that parents of children with disability were particularly fearful of the COVID-19 vaccinations, due to a lack of information about potential negative side effects. This fear was reported to impact the uptake of other vaccines, as a female disability worker commented:

The vaccination for COVID-19 is what scared a lot of people because there was not as much information that was going out for children with disabilities and the effects it would have on them. [. . .] There was a lot of, you know, miscommunication or difficulty in under-standing or just a lack of information and scared to get vaccination [Female, Disability worker].

Participants believed that if parents were better informed and aware, they would be more inclined to vaccinate their children with disability. Participants explained that advocating to parents about the importance of vaccination was essential to build bridges to vaccine acceptability, "the awareness should always be out there for the parents on the importance of vaccines because it's their right. Once they know about. . . once they are aware of the knowledge and the importance of the vaccines, they will come" [Female, Community/ faith leader]. To achieve this, participants explained, "We need clear information shared, and it needs to be adapted to suit all individuals." This included "information to suit all types of impairment", including "in pictorial, or in simple terminology" [Female, Disability worker].

## Social processes

**"What's wrong with your child?": Disability stigma and discrimination.** Participants reported that children with disability and their parents experienced stigma and discrimination when accessing healthcare facilities and that this reduced engagement with vaccination programs, "we see the shame the parents have when bringing their disabled children to the health centers" [Female, Community/ faith leader]. Issues such as being stared at and asked invasive questions by other patients meant that "the parents are not too keen to come to the normal clinics [local health center]" [Female, Health worker].

[There is a] certain stigma that the parent or the carer goes through coming into the healthcare facility, umm, stigma from the way people. . . the way the general public perceives. . . you know, the staring, the looking, sometimes even the asking of questions, like "What's wrong with your child?", and that sometimes can burden parents into, you know, only coming into the healthcare facility if the child is very unwell. Something routine like getting a vaccination may not be as much of a priority [Female, Health worker].

One participant emphasized the challenges faced by parents of children with disability, noting that dealing with societal stigma was often more difficult than coping with the child's impairments, "they [parents of children with disability] usually say it's much more difficult to deal with the stigma then dealing with the nature of diagnosis of living with that disability" [Female, Disability worker]. Participants also observed that "because they are always being discriminated against, they're always being left behind" [Female, Disability worker] and that the families of children with disability often preferred "to be on their own or isolate themselves rather than accessing public services or benefitting from all the assistance that's currently in place" [Female, Disability worker].

**One size doesn't fit all: Need for tailored communication and engagement initiatives.** Participants identified the need for a range of tailored initiatives to build awareness and understanding among parents of children with disability about the need for vaccination. Participants said it was necessary to "increase the knowledge in the community, making them understand why their child needs to receive this vaccine" [Female, Health worker] and that "working with them at their level" [Female, Health worker] was crucial. Participants explained that engaging parents of children with disability demanded a thoughtful approach including collaboration with local leaders, "whether you use the church as a platform or you use the village setting, you know the Turaga-ni-koro or the Nasi-ni-koro, all these different platforms within the community are useful when reaching out to this population" as "they will listen to people that they trust already" and "we just do not want to leave any disabled child behind and not vaccinated for whatever the reason" [Female, International Agency]. Community champions also

included "individuals respected within their communities" such as "a teacher or doctor in that community or it's their pastor" [Female, International Agency]

> So we must identify trusted people that they relate to, like, if it's somebody who has many children already and the children are successful, and she talks about how they are all vaccinated and because of that they are all healthy and working well [Female, International Agency].

Participants explained that there was a need to tailor vaccine information to the different communication needs of community members as "different contexts require different content:" [Female, Disability worker].

> When it comes to a person with disability, we normally say one size does not fit all meaning the information or the communication you share to a person who is blind or with low vision will not be the same as with a person who has psychosocial impairments or as to a person who has down syndrome or in fact a deaf person [Female, Disability worker].

"Promoting [vaccinations] using social media or TV" [Female, Health worker] was also identified as a powerful platform to provide information to parents of children with disability. Participants suggested that these methods could have far-reaching impacts, especially when compared to traditional methods like pamphlets and posters:

> Having the correct information that's widely disseminated and since now a lot of parents have access to smartphones, it's accessible rather than having it, for instance, on posters in public places. I feel that more and more people respond to social media campaigns rather than typical pamphlets or posters; it's more widely acceptable [Female, Health worker].

However, participants emphasized that it was important to "ensure that information is correct using local terms or in the local vernacular" [Female, Disability worker].

**Need for improved disability and health service collaboration.** A key factor that stood out during focus groups was the need for increased collaboration between disability and health services, so that "everybody's load is shared" [Female, Disability worker]. Participants suggested that disability services were often a trusted source of information and could be instrumental in encouraging vaccination:

> For us as an association for people with disability, we see that they are more confident to access things when it comes from us. When they have questions, they confidently ask as well, and when we need more information, we can refer back to help us understand more, then we get back to them [Female, Disability worker].

Disability services also played an important role in directly advocating to relevant health authorities to ensure that when children with disability attend clinics, they were identified as requiring vaccination, "we do look at the MCH (Maternal and Child Health) card, and that's where we can advocate for the children, you know, umm 'you have missed out on these vaccinations and if you can have them, update them'" [Male, Disability worker] and "what we can definitely do is highlight to the Ministry of Health and Colonial War Memorial Hospital (CWMH), 'look here, these children have been missing out, when they do come for clinics, can you have them take the immunizations'" [Female, Disability worker]. Community health

workers were essential in this communication as they provided continuity of continuum of care and could directly advocate to parents for participation in vaccination programs:

> It's important to include community health workers in this process. This will help what I think is from the point of diagnosis. Because once a diagnosis is made, the pathway should be clear, and communication from hospital to health center, to nursing station. That channel of communication is important because that's how we are going to be able to follow up and provide further assistance if needed for the family [Male, Health worker].

However, participants commented that "often families move to other areas and do not inform the health centers" [Female, Health worker], which resulted in loss to follow-up. To address this, a participant suggested that "re-introducing community rehabilitative system assistance" [Female, Health worker] would be ideal as this method had worked in the past and would ensure that all children are covered in terms of services necessary for their wellbeing:

> One of the things we used to have before was community rehabilitative system assistance [. . .] it was very valuable [. . .] When you refer them to a different site or place, we don't get any feedback on whether they really follow through with what we ask of them but if . . . if there's a one-stop shop [. . .] if they access a health facility, that's well and good and we'll continue that communication and follow through until the children are vaccinated working with primary school with the school health team [Female, Health worker].

These findings emphasize the importance of disability and health services working together to improve vaccine uptake for children with disability.

## Motivation

**Awareness, support, and religious barriers impacted vaccine motivation.** Participants reported that a minority of parents express a desire for their children with disability to be vaccinated, "the strength is there, they want their child to be vaccinated" [Female, Health worker]. More commonly, participants explained that parents, particularly mothers as "hardly you'll see fathers bringing the child" [Male, Health worker], lacked support and faced structural barriers in caring for their children with disability, preventing engagement with vaccination:

> Mothers are so tired taking care of their disabled children, and there is no other caregiver, there are no health workers, no help, so they just neglect the children and focus on the other children [Male, Community health worker].

Parent's lack of access to accessible information about healthcare services for their children with disability was also cited as a factor contributing to low vaccination rates:

> Not knowing that even though their child has a disability that still means that the child can access the healthcare system for vaccinations, just like any other child who does not have a disability, that's one barrier [Female, Health worker].

Additionally, religious beliefs complicated vaccination engagement. Participants explained that vaccination "all depends on the family agreeing or disagreeing" and that vaccine hesitancy was most common among parents who were "faith-based" [Female, Community/ faith leader]:

For some, it was about the church that they follow, that [vaccination] was not encouraged, or they felt that they had other ways that they could deal with protecting their children, whether it's through diet or other traditional ways to protect their children [Female, International Agency].

These findings emphasize the numerous factors that impact parental motivation regarding vaccination for children with disability.

## Practical issues

**Access issues: Lack of disability-friendly environments, financial and time barriers.** Children with disability face numerous barriers to accessing vaccination services, according to participants. For instance, health settings could be unsettling for children with disability, "a hospital or a clinical setting can be very traumatic. So, you know there can be a lot of tantrums, and a very traumatic situation can trigger some not-so-pleasant behavior [Female, Disability worker]. Additionally, participants explained that at some clinics, there was nowhere for children with a physical disability to wait comfortably, "the child who needs to be carried, like there's lots of stiffness and the parent has to be having to settle the child and, in that sense, in having to carry the child is not a place for the child to be put down and wait" [Female, Health worker].

Participants reported that for many families of children with disability, particularly low-income families living remotely, the non-dose related costs and time required to travel to vaccination appointments were prohibitive, "a thing that hinders them is their geographical location in terms of the nearest health facility and uhhh, transportation, the mode of transport [. . .] if they have to come in public transport then it's a difficult task" [Male, Health worker].

Participants also explained that children with disability often required longer appointment times, causing difficulty for low-income earners:

It becomes difficult for working parents to take time off to take their child for vaccination, but when you have a child with behavioral concerns, a 20-minute vaccination, um, just 20 minutes to get the child vaccinated, could turn into 2 hours [Female, Disability worker].

To mitigate these issues, participants suggested there was a need to create disability-friendly environment in health facilities:

One of the ways in which you can make vaccination services more accommodating to children with disability or behavior concerns is by setting up a process that allows children with disabilities to go first instead of having to wait for their turn because once they hear other children screaming or crying, you're never going to be able to get that child vaccinated [Female, Disability worker].

Outreach services were also recommended, "bringing the services closer to them helps the coverage and the uptake" [Female, International Agency]. However, it was noted that families of children with disability "have 101 things to do- and sometimes health is not the priority for them" so it is important to "align our outreach programs to suit their needs" [Female, International Agency].

Even if we visit them in the daytime, it doesn't work, the house is empty and so forth, the whole family has gone to the farm. But uhm, encouraging the health facilities to align their outreach program to work with the community, what time works best, or they have a

dedicated time uhm for that month that they will visit that community, so they're prepared and not go to the farm [Female, International Agency].

Reinstating reminders for parents who have a tough time keeping up with their children's vaccination schedules and conducting community vaccination once a month were suggested to be beneficial ways to engage children with disability in hard-to-reach locations. Participants explained that accessing health services "should be enabling for the parents and children living with disability and environmentally friendly, as this gives the parents that boost to know that their children are special and there's no limitation" [Female, Health worker].

**"That trust is kind of gone": Lack of knowledge, confidence, and informed consent administering vaccines.** Participants reported a lack of knowledge and confidence in providing vaccinations to children with disability. For instance, "nurses are not confident in giving [the vaccination], like, we don't know the kind of complications that may arise" [Female, Health worker]. Participants explained that for this reason, "we need to strengthen, more training, and more awareness on how to vaccinate our disabled children" [Female, Health worker]. The need to "educate healthcare workers to be knowledgeable on what a child with disability really encompasses" [Female, Health worker] was emphasized, including "highlighting that using the proper terms so that everyone is on board" [Female, Health worker]. Some healthcare workers also lacked up-to-date knowledge about vaccinations, due to staff turnover and new vaccination rollouts:

New vaccines will become available, new formulations, you know. This will need to be informed to the healthcare workers, so they know what they are dealing with before they start giving it to the communities [Female, International Agency].

A number of issues communicating with parents of children with disability about vaccines were identified. Several participants reported instances of healthcare workers administering vaccinations to children with disability without first obtaining parental consent or explaining the nature of the vaccinations. These experiences made parents hesitant to visit medical centers out of fear their children would receive vaccinations without their consent:

The nurse just picked up the needle and injected the little kid without explaining what the vax was for. That will make the parents fearful in future when there is a call for immunization, you know . . . because nothing has been explained to them. They don't want any foreign material to come into their children [Female, Community/ faith leader].

In a small number of cases, children with disability had "gotten the wrong vaccination" [Male, Disability worker]. A participant explained, "there is a need to improve health services to remove those human errors that are more prominent and come up with some ways that we can keep these errors from happening often" [Male, Disability worker].

Participants from some health services said their organizations were revising policies to be more inclusive of children with disability, "we are revising the immunization policy with a separate SOP [Standard of Procedure] and looking at inclusiveness to include this vulnerable population of disabled children and how they are managed" [Female, Health worker]. Participants explained that "once the SOP is amended, the healthcare worker will conduct awareness, policy awareness and training" as "it's very important that the healthcare workers know their roles, they need to play their role well" [Female, Health worker].

## Discussion

The findings of this study provide insight into the multifaceted psychological, social, cultural, and environmental barriers to vaccination for children with disability in Fiji from the perspectives of health, disability, and community stakeholders. We found that the vaccination of children with disability in Fiji was hindered by a lack of information and support for parents of children with disability, concerns about vaccine benefits and safety, healthcare workers' lack of confidence and knowledge in administering vaccines to children with disability, understanding vaccine policies and schedules for children with disability and communication and trust issues with healthcare workers. Additionally, children with disability were reported to encounter challenges accessing vaccinations due to issues such as long waiting times for appointments, inadequate waiting areas, financial and time barriers, and disability stigma and discrimination within healthcare settings.

The relationship between vaccine uptake and safety concerns is well established in general vaccine literature [28]. These concerns may be exacerbated for families of children with disability due to pervasive myths incorrectly linking vaccinations with childhood disability [29]. Previous research showed that healthcare workers can play a vital role in addressing these concerns by communicating the evidence of strong scientific agreement supporting vaccines [30, 31], debunking vaccine myths with humor [30], and warning people about the potential for misinformation [30]. However, as this study identified, healthcare workers in Fiji lacked the knowledge and confidence to answer parents' questions and alleviate concerns about vaccination for children with disability. Poor communicative competence about disability and inadequate informed consent procedures were also evident, impacting trust between patients and providers. Similar findings were reported from a recent study in Indonesia, where a lack of confidence to answer parent's questions was associated with stress and trauma [32]. These findings highlight the need to strengthen the health workforce capacity and capability in an ongoing manner. However, this poses a significant challenge in the study setting due to difficulties in retaining skilled healthcare workers in Fiji [33]. In 2022, for instance, more than a quarter of all trained nurses were estimated to have migrated overseas, contributing significant skills and knowledge loss within the health system [33]. System and policy responses are needed [34], including funding prioritization for workforce growth, providing training about vaccine safety for children with disability, and improving communication practices to enhance informed consent processes. These steps will help create a more supportive environment for vaccination in this population [19].

In Fiji, like many other countries around the world, school-based vaccination (SBV) and checking students' vaccination records at school are used to optimize vaccination coverage [35]. However, children with disability in Fiji are less likely to attend school than children without disability and may be missed through these methods [36]. Disability service providers, some of whom are education providers [37], are often well connected to this population and are trusted providers of health information. This makes them a valuable resource to challenge misinformation and encourage vaccination. By capitalizing on the effectiveness of this local, interpersonal communication [38], future vaccination campaigns can benefit from working collaboratively with disability service providers [39] and other community champions [40] to support decision-making. Additionally, tailored public education and awareness activities for children with disability and coordinated advertising campaigns across various media technologies have the potential to reach wider audiences within this population [41]. This multi-pronged approach will maximize the impact of future vaccination campaigns for children with disability.

The provision of accessible, safe, and inclusive vaccination services for children with disability is a critical step towards ensuring equity of access [5]. However, nearly a quarter of

children with disability (23.1%) in our previous research reported difficulties accessing vaccinations [16] and a number of barriers to access were identified. In addition to improving the accessibility of vaccination clinics, alternative community-led solutions for vaccination [42], such as the (re)implementation of outreach programs [43], house-to-house vaccinations [44], and after-hours vaccination clinics are necessary to address uptake barriers. Parental reminder strategies are also demonstrated to significantly increase immunization uptake, such as mobile text message reminders [45] and may help alleviate barriers to vaccination identified in the present study. To ensure effectiveness and sustainability, such initiatives should prioritize accessibility, cultural sensitivity, and community engagement [5]. Vaccination is a cost-effective investment for governments, even when additional expenses for addressing accessibility barriers are calculated [46].

Our study had many strengths. It is the first in Fiji and other Pacific Island countries to investigate drivers and barriers to childhood disability vaccination. We collaborated with lead health and disability service providers and researchers in the region in the study design, implementation, and analysis, and the inclusion of multiple stakeholder perspectives. These approaches ensured the relevance and applicability of findings to facilitate immediate adoption into practice. Throughout the research, our team engaged in continuous self-reflection, critically considering how our identities and experiences influenced our interpretation of the data. For instance, our team's diverse cultural and disciplinary backgrounds enriched the analysis by providing culturally relevant insights and avoiding Western-centric misinterpretations. Team members with experience in Fiji and other low- and middle-income settings brought essential understanding and considerations for local health practices and community attitudes toward disability and vaccination. Disability researchers contributed to deep awareness of the barriers faced by people with disability, while vaccination experts provided valuable perspectives on public health strategies and immunization systems. This reflexivity guided our decisions on setting boundaries for inclusion within individual themes, enabling us to identify and generate "themes-as-shared-meaning" [27, p.342], leading to a rich and nuanced analysis.

This study has several limitations. First, our focus was specifically on immunization and disability stakeholders which meant we did not collect qualitative data from children with disability and their families. While this approach allowed us to gather insights directly from those influencing vaccination policies and practices, it limits our understanding of the experiences and perspectives of the children and families most affected by these issues. The focus groups were also small, which may restrict the diversity of viewpoints captured. Fiji also has a small workforce [18] which meant that there are only a small number of experts and stakeholders who can be consulted. Furthermore, while our research examined vaccine uptake and social and behavioral drivers using the WHO BeSD survey tools, future research should examine effective interventions to increase vaccine uptake in this population. These approaches should be co-designed with children with disability and their families to address barriers to vaccination specific to children with disability in LMICs [28].

## Conclusions

In conclusion, our findings demonstrate the numerous and complex drivers and barriers to vaccination faced by children with disability in Fiji, highlighting the urgent need to enhance the inclusiveness of vaccination health services and communication for this population. This includes promoting collaboration between disability and health services to optimize and strengthen immunization referral pathways and providing education and training for healthcare workers to build capacity in working with children with disability and their families. Training should encompass strategies to address disability stigma, contraindications, if any,

including vaccine safety, and clear protocols for communication and consent tailored to children with disability and their families. Additionally, training should equip healthcare workers with practical skills for adapting vaccination approaches to support children with varying mobility and communication needs. Additionally, tailored public education and awareness activities for children with disability, coordinated advertising campaigns across various media technologies, improved accessibility of vaccination clinics, and the (re)implementation of outreach programs, house-to-house vaccinations, and after-hours vaccination clinics are necessary to improve vaccine uptake. By understanding and addressing the behavioral and social factors contributing to the low vaccination rates of children with disability, targeted public health interventions and vaccination campaigns can be developed. These efforts will empower families to vaccinate their children with disability and improve the knowledge and confidence of health workers to administer vaccines to this population.

## Supporting information

**S1 Checklist. Inclusivity in global research checklist.**
(DOCX)

**S1 Appendix. Focus group interview guides for immunization healthcare workers and for disability and community stakeholders.**
(DOCX)

## Acknowledgments

We thank all study participants for their time and information. Sincere thanks to the team members of the Ministry of Health & Medical Services, Colonial War Memorial Hospital, Frank Hilton Organization, and Fiji National University for their immense support in implementing the project throughout the study period. We thank Litiana Seru, Research Assistant, Fiji National University for her involvement in transcription and integrity checking. We thank Susana Lolohea from the Fiji National University for her support with project administration and logistics of study implementation.

## Australian immunisation and disability investigators involved in this paper

Susan Woolfenden[a]; Margie Danchin[b]; Sarah McIntyre[c]; Hayley Smithers-Sheedy[c]; Nadia Badawi[a,c]; Kristine Macartney[a,d]

[a] Sydney Medical School, the Faculty of Medicine and Health, The University of Sydney, New South Wales, Australia

[b] Murdoch Children's Research Institute and University of Melbourne, Royal Children's Hospital, Parkville, Victoria, Australia

[c] Cerebral Palsy Alliance, New South Wales, Australia

[d] National Centre for Immunisation Research and Surveillance (NCIRS), New South Wales, Australia

## Author Contributions

**Conceptualization:** Unise Vakaloloma, Israt Jahan, Rachel Devi, Meru Sheel.

**Data curation:** Unise Vakaloloma.

**Formal analysis:** Rosalie Power, Unise Vakaloloma, Israt Jahan, Sureni Perera, Ilisapeci Tuibeqa, Litiana Volavola, William May, Donald Wilson, Lanieta Tuimabu, Gulam Khandaker, Meru Sheel.

**Funding acquisition:** Meru Sheel.

**Investigation:** Unise Vakaloloma, Israt Jahan, Meru Sheel.

**Methodology:** Israt Jahan, Sureni Perera, Ilisapeci Tuibeqa, Rachel Devi, Litiana Volavola, William May, Donald Wilson, Lanieta Tuimabu, Gulam Khandaker, Meru Sheel.

**Project administration:** Unise Vakaloloma, Meru Sheel.

**Supervision:** Meru Sheel.

**Writing – original draft:** Rosalie Power, Unise Vakaloloma, Israt Jahan, Meru Sheel.

**Writing – review & editing:** Rosalie Power, Unise Vakaloloma, Israt Jahan, Sureni Perera, Ilisapeci Tuibeqa, Rachel Devi, Litiana Volavola, William May, Donald Wilson, Lanieta Tuimabu, Gulam Khandaker, Meru Sheel.

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
