## [Decision Letter · Decision Letter 0]

20 Oct 2024

PGPH-D-24-01875

"Fear of the unknown": Health, disability, and stakeholder perspectives on the behavioral and social drivers of vaccination in children with disability in Fiji.

Dear Dr. Sheel,

Thank you for submitting your manuscript to PLOS Global Public Health. After careful consideration, we feel that it has merit but does not fully meet PLOS Global Public Health’s publication criteria as it currently stands. Therefore, we invite you to submit a revised version of the manuscript that addresses the points raised during the review process.

We have benefited from receiving several reviewers. Most of these have very positive comments towards your manuscript. I expect you to respond to all reviewer comments, but I understand that it may be difficult to edit the manuscript in response to every single query - so it is okay to explain to me and the reviewers if you decide to not make a particular change. Note that two reviewers provided comments in separate Word documents.

We look forward to receiving your revised manuscript.

Kind regards,

Abram L. Wagner, PhD, MPH

Academic Editor

Journal Requirements:

2. Please provide separate figure files in .tif or .eps format.

Additional Editor Comments (if provided):

Reviewers' comments:

Reviewer's Responses to Questions

**Comments to the Author**

1. Does this manuscript meet PLOS Global Public Health’s publication criteria? Is the manuscript technically sound, and do the data support the conclusions? The manuscript must describe methodologically and ethically rigorous research with conclusions that are appropriately drawn based on the data presented.

Reviewer #1: Partly

Reviewer #2: Yes

Reviewer #3: Yes

Reviewer #4: Yes

Reviewer #5: Yes

Reviewer #6: Yes

Reviewer #7: Yes

2. Has the statistical analysis been performed appropriately and rigorously?

Reviewer #1: Yes

Reviewer #2: N/A

Reviewer #3: Yes

Reviewer #4: I don't know

Reviewer #5: N/A

Reviewer #6: N/A

Reviewer #7: N/A

3. Have the authors made all data underlying the findings in their manuscript fully available (please refer to the Data Availability Statement at the start of the manuscript PDF file)?

Reviewer #1: Yes

Reviewer #2: Yes

Reviewer #3: Yes

Reviewer #4: Yes

Reviewer #5: Yes

Reviewer #6: Yes

Reviewer #7: Yes

4. Is the manuscript presented in an intelligible fashion and written in standard English?

Reviewer #1: Yes

Reviewer #2: Yes

Reviewer #3: Yes

Reviewer #4: Yes

Reviewer #5: Yes

Reviewer #6: Yes

Reviewer #7: Yes

5. Review Comments to the Author

Reviewer #1: TOPIC

"Fear of the unknown": Health, disability, and stakeholder perspectives on the behavioral and social drivers of vaccination in children with disability in Fiji.

Remark(s) 1: Remove the full stop from the topic, otherwise, it reads fine.

ABSTRACT

Background:

Page 3, sentence 2, lines 33-35: Understanding the attitudes and behaviors of health, disability, and community stakeholders toward vaccination uptake in these children is important, as they can impact the success of vaccination programs.

Remark(s) 2: Recast to read: “…uptake among children is important, as they can impact the success of vaccination programs.”

Remark(s) 3: Remove all abbreviations from the abstract, there is no need for them.

Conclusions:

Page 3, lines 50-53: The findings from this study can guide strategic action….

Remark(s) 4: This is not a good conclusion, recast.

INTRODUCTION

Page 5, para2, lines 84-87: To better understand access to immunization among children with disability in Fiji and to inform the present study, we conducted a cross-sectional survey to examine vaccination rates in children with disability. We found that only 55% of children with disability in our study location were vaccinated against basic antigens.16

Remark(s) 5: This portion is problematic. Did the authors conduct this previous study in furtherance of the current study? This portion should be reported under the methods given the link with the current study. I surmise the authors are trying to build a strong justification for the current study, however, the presentation is not clear. I suggest a complete rephrase of this portion of the problem statement.

MATERIALS AND METHODS

Study location:

Page 5, lines 101 & 102: This study was conducted in peri-urban and urban areas of the Suva–Nausori corridor in Fiji.

Remark(s) 6: This is inadequate in providing details on the location.

Design and theoretical framework

Page 6, lines 108-113: We used the World Health Organization’s (WHO) Behavioral and Social Drivers (BeSD) of Vaccination Framework20. A 12-member stakeholder group was actively engaged and involved throughout all stages of the project providing expertise to guide the study design, data collection, analysis, and dissemination of results. The group included local and international disability and health service providers (study partners), MHMS, academics, and community and faith leaders.

Remark(s) 7: There are no details on the design and theoretical framework provided; reorganize this sub-section. This can undermine the reproducibility of the study. Moreover, what is the justification for the choice of design? You need to justify this with appropriate sources.

Recruitment and participants:

Page 6, lines 115-128: Immunization health workers, disability service providers, and advocates from local non-government….

Remark(s) 8: There are issues with this sub-section. First, there was no source to support the sampling technique. Second, what is the size of the target population? What are the exclusion criteria?

Procedure:

Page 7, lines 131-134: Two focus group interview guides (one for immunization healthcare workers and another for disability and community stakeholders) were adapted from the WHO BeSD childhood vaccination qualitative framework analysis template tool….

Remark(s) 9: If the interview guide was adapted, then you need to provide details on what was added or removed or varied in the WHO BeSD tool. Meanwhile, you reported earlier under the design and theoretical framework that you used the World Health Organization’s (WHO) Behavioral and Social Drivers (BeSD) of Vaccination Framework. You must be consistent on how the instrument was developed.

Data analysis

Page 8, lines 154-174: A combination of inductive and deductive reflexive thematic analysis was ….

Remark(s) 10: How did you ensure qualitative rigour?

RESULTS

Remark(s) 11: While the section provided a detailed account of the themes that emerged from the data analysis, the authors failed to show whether the findings were expected or not. At least, the authors must go beyond the mere presentation of the findings and provide their own assessment of same by clearly demonstrating whether the findings were surprising or not.

LIMITATIONS

Remark(s) 12: Provide a section for limitations of the study

COMMENTS FOR AUTHORS

Generally, the authors did well in exploring an area of public health that is yet to receive the needed research attention. Overall, the paper was well written and the authors have demonstrated a good knowledge of the subject matter. However, there are observations needing proper resolution by the authors to improve the quality of the paper and make it more beneficial for the global readership. Specific comments on observations made have been provided above.

Reviewer #2: A well written manuscript. The results adequately support the authors' claims regarding the challenges faced by disabled children in Fiji regarding their right to access to care.

The methodology, which combines a cross-sectional survey with focus groups provides a valuable mixed methods design of both vaccination rates and stakeholder's perspectives. The sampling strategy is appropriate. The use of the WHO BeSD Framework provides a strong theoretical underpinning for the study and the iterative process of thematic analysis by a number of researchers, enhances the reliability and validity of the findings. More attention could be given to the information provided about the potential biases and their mitigation in both data collection and analysis. The recommendations offer practical implications for health policy improvements for this population and provide ideas for actionable interventions. In summary, this manuscript presents a well-structured study that contributes valuable knowledge to understanding vaccination disparities among children with disabilities in Fiji. The results are compelling, methodologies sound, ethical considerations robust, and conclusions appropriately derived from the findings.

Reviewer #3: The manuscript entitled "Fear of the unknown": Health, disability, and stakeholder perspectives on the behavioral and social drivers of vaccination in children with disability in Fiji" is an essential study for vulnerable groups, and the research topic is critical as all children with disabilities should be vaccinated and prevented from vaccine-preventable diseases. The study presents important findings through a qualitative study using five focus group discussions on what hinders children with disabilities from getting vaccinated. They found that stigma, low knowledge of the benefits of vaccines, misinformation, and inaccessible health service delivery are the factors that hinder children with a disability from getting vaccinated.

The paper is well written except for a few typo errors, such as on Page 5, Line 88, the word "knows" should be "known," and on Page 6, Line 105, there should be a period to separate two sentences.

I am only concerned about the use of FGD as a data collection technique, where the number of participants in some FGDs was less than four. Hence, justifying the need for FGD for this study would improve the paper. Similarly, I could not see much text on the trustworthiness of the findings. I think explaining about it could benefit the paper.

Overall, the paper is fantastic and easy to read. Thank you.

Reviewer #4: The manuscript offers a valuable contribution to the understanding of the behavioural and social factors influencing vaccine uptake among children with disabilities in Fiji, as perceived by key stakeholders. It is emphasised that there is a necessity to diminish the prevalence of preventable illnesses among disabled children in Fiji. Nevertheless, a number of points require further clarification and refinement :

1. Abstract :

While the study offers valuable insights, However, the abstract is not sufficiently precise. It is recommended that the authors refine the writing of these two subsections, namely 'Context' and 'Methods'.

For instance, the qualitative research approach employed in this study should be explicitly delineated in the 'methodology' subsection, while the study's objective should be clearly articulated in the 'context' subsection.

2. Methods :

It is recommended that the authors limit the scope of the study to the essential elements.

Furthermore, the authors should provide clarification regarding the qualitative research approach employed. The text could be deleted or made more fluid by replacing it with the following: 'The study consisted of two parts:

'A) A survey examining vaccination among 198 disabled children and their carers; B) focus groups with health, disability and community stakeholders on behavioural and social factors of vaccination among disabled children.

This is because the authors state in the 'Introduction' section that they conducted an initial quantitative study (cf. last paragraph of the 'Introduction' section, from 96-101).

It is therefore difficult to understand the design of the current study.

Reviewer #5: This study sheds light on the often-overlooked barriers to vaccination for children with disabilities in Fiji. The title reflects the key themes, but it is too long.

The introduction highlights the importance of vaccination in reducing childhood morbidity and mortality but could be more concise, especially in discussing vaccination campaigns.

The term "intersectional inequality" is not fully unpacked for readers unfamiliar with its complexities, especially in the specific context of disabilities and vaccination.

The recruitment process is clear, but clarify the roles between different groups (e.g., health workers, disability providers, and advocates) by specifying their unique roles in the study.

The mention of managing power dynamics is good, but explain how grouping by hierarchical position achieves this.

The procedure section is detailed. The process of adapting the interview guides from the WHO BeSD framework could be better explained to show how local relevance was incorporated.

In-depth probing was improved after the pilot test, but more detail on the specific changes made would add clarity.

Rapport-building and using body language to encourage participation are useful, but this part could be more concise.

The data analysis section is thorough. Are there specific examples of how the researchers' cultural backgrounds and personal experiences influenced the coding and analysis process? Currently, the mention of reflexivity feels superficial.

The findings highlight the challenges of vaccinating children with disabilities. While the quotes add depth, some are repetitive.

A key issue is healthcare workers' lack of confidence and knowledge in administering vaccines to this group, and more details on the necessary training would help.

The informed consent issues, where vaccines are given without proper explanation, need stronger focus on improving communication.

The mention of policy changes to include children with disabilities is promising, but more specifics on how these will be implemented would add valuable context.

The study highlights the psychological, social, cultural, and environmental barriers to vaccinating children with disabilities in Fiji. While the connection to global vaccine uptake literature is relevant, there could be more emphasis on actionable steps to improve healthcare workers’ communication and trust-building with families.

The role of disability service providers in promoting vaccination is well-supported, but the discussion around solutions like outreach programs and after-hours clinics could be more concise.

The study’s strengths, such as stakeholder involvement and reflexivity, are clear, though future directions could be more succinct.

Overall, the discussion is comprehensive but would benefit from sharper focus on practical implications.

There are some minor grammatical errors in the paper, and the use of the terms ‘disability’ and ‘disabilities’ is not consistent.

Reviewer #6: • The authors should provide more details on the study area Suva–Nausori corridor in Fiji. Is this in the south of the country or the north or east? Is this a State in the country? This will help readers understand the nature of the context.

• How many participants were in each of the focus groups (5) since they have 22 participants in total?

• Global literature on conducting an FGD suggests a minimum of 5-6 participants and a maximum number of 12 participants?

• How was the participants allocated to each FGD determined?

• Do the investigators think the number in each FGD is adequate?

• The investigators should provide the codebook that was used in the content and thematic analysis process as a supplementary file?

• Table 1 should be added to the supplementary files.

Reviewer #7: This study conducted in the context of low and middle income country is one of its first kind to be studied and therefore has merit and add to the body of knowledge and have implications for policy planning.

See attachment for my comments and feedback.

6. PLOS authors have the option to publish the peer review history of their article (what does this mean?). If published, this will include your full peer review and any attached files.

**Do you want your identity to be public for this peer review?** For information about this choice, including consent withdrawal, please see our Privacy Policy.

Reviewer #1: **Yes: **BOTHA Nkosi Nkosi

Reviewer #2: No

Reviewer #3: No

Reviewer #4: **Yes: **Chadrack KABEYA DIYOKA

Reviewer #5: No

Reviewer #6: **Yes: **Saheed Akinmayowa Lawal

Reviewer #7: No

---

## [Editor Report · Decision Letter 1]

16 Dec 2024

"Fear of the unknown": Health, disability, and stakeholder perspectives on the behavioral and social drivers of vaccination in children with disability in Fiji

PGPH-D-24-01875R1

Dear Dr. Sheel,

We are pleased to inform you that your manuscript '"Fear of the unknown": Health, disability, and stakeholder perspectives on the behavioral and social drivers of vaccination in children with disability in Fiji' has been provisionally accepted for publication in PLOS Global Public Health.

Best regards,

Abram L. Wagner, PhD, MPH

Academic Editor
